# Influence of Multiple Anti-VEGF Injections on Retinal Nerve Fiber Layer and Ganglion Cell-Inner Plexiform Layer Thickness in Patients with Exudative Age-Related Macular Degeneration

**DOI:** 10.3390/medicina59010138

**Published:** 2023-01-10

**Authors:** Maja Zivkovic, Aleksandra Radosavljevic, Marko Zlatanovic, Vesna Jaksic, Sofija Davidovic, Miroslav Stamenkovic, Ivana Todorovic, Jana Jaksic

**Affiliations:** 1Department of Ophthalmology, Medical Faculty, University of Nis, 18000 Nis, Serbia; 2Ophthalmology Eye Hospital “Clinic Maja”, 18000 Nis, Serbia; 3School of Medicine, University of Belgrade, 11000 Belgrade, Serbia; 4Eye Clinic, University Clinical Center of Serbia, 11000 Belgrade, Serbia; 5Eye Clinic, Clinical Center Zvezdara, 11000 Belgrade, Serbia; 6Department of Ophthalmology, Medical Faculty, University of Novi Sad, 21000 Novi Sad, Serbia

**Keywords:** anti-VEGF, exudative age-related macular degeneration, spectral domain optical coherence tomography

## Abstract

*Backgrounds and Objectives*: To analyze the influence of multiple anti-VEGF intravitreal injections for exudative age-related macular degeneration on the thickness of peripapillary retinal nerve fiber layer (RNFL) and macular ganglion cell-inner plexiform layer (GC + IPL) using spectral domain optical coherence tomography (SD-OCT). *Materials and Methods*: A prospective interventional study of consecutive patients treated with intravitreal bevacizumab (IVB) was performed. Average and sectorial values of RNFL and GC + IPL thickness were recorded using Cirrus SD-OCT at 0, 6, 12, and 24 months. Patients suffering from any ocular disease that could affect RNFL or GC + IPL thickness were excluded. *Results*: A total of 135 patients (70 women and 65 men, aged 65 ± 15 years) were included. The average number of injections per patient was 12.4 ± 2.4. Average RNFL and GC + IPL thickness prior to the first injection (87.6 ± 12.2 and 47.2 ± 15.5 respectively), and after 24-month follow-up (86.2 ± 12.6 and 46.7 ± 11.9 respectively) did not differ significantly (*p* > 0.05). There was a significant decrease in GC2, GC5 segments, and minimum GC + IPL thickness. *Conclusion*: Repeated anti-VEGF treatment did not cause significant changes in the thickness of RNFL and GC + IPL layers over a period of 24 months. The detected decrease in GC2 and GC5 sectors, as well as in minimum GC + IPL thickness, could be a sign of ganglion cell damage induced by the treatment or could occur during the natural course of the disease.

## 1. Introduction

Age-related macular degeneration (AMD) is considered to be one of the leading causes of visual impairment among developed countries. However, the advent of anti-vascular endothelial growth factor (anti-VEGF) therapy led to a huge shift in treatment of exudative form of this disease. Thanks to this therapeutic option, it is possible—in a carefully selected group of patients—to prevent disease progression and even significantly improve visual acuity. However, long-term side effects of intravitreal application of anti-angiogenic factors are still under investigation in large clinical trials worldwide.

Endogenous VEGF has been shown to have an important role and particular significance in the maintenance and function of the retina. It has been involved in numerous vital processes important for adult retinal function, such as angiogenesis, regulation of microvascular tone and permeability, inflammatory responses, neuroprotection in response to retinal ischemia, neurotrophic effects, and regulation of cell proliferation [1,2]. Therefore, repeated injections of anti-VEGF agents should be administered with caution since they may result in long-term suppression of endogenous VEGF that could cause many long-term and potentially harmful side effects on neuronal cells. Recently published papers have revealed that anti-VEGF therapy could cause changes in choriocapillaris endothelial cell fenestration, retinal arteriolar diameter, choroidal thickness, [3] and—finally—atrophy of the retinal pigment epithelium (RPE). Increase in intraocular pressure following intravitreal anti-VEGF injections could potentially have harmful effects on the neuronal cells. In several studies, intravitreal injection of anti-VEGF agents resulted in significant transient increase in intraocular pressure (IOP) that never occluded central retinal artery and spontaneously diminished by 30 min post injection [4,5]. This may be damaging to the optic nerve, particularly in patients with advanced glaucoma [6]. On the other hand, longer duration of ocular hypertension could be a significant problem and was reported in several studies [7,8,9,10]. Aqueous outflow can probably be compromised by accumulated material in trabecular meshwork and/or inflammatory response within it [10]. However, other large retrospective cohort studies did not find significant IOP elevation after multiple anti-VEGF intravitreal injections in AMD patients [11,12].

There is evidence that retinal nerve fiber layer (RNFL) and ganglion cell complex (GCC) thicknesses may decrease after anti-VEGF therapy due to lack of neuroprotective effect of VEGF on RNFL and/or temporary or permanent IOP elevation after multiple intravitreal injections. The aim of this study was to analyze the changes in the RNFL and ganglion cell-inner plexiform layer (GC + IPL) thickness in patients with exudative form of AMD after multiple intravitreal injections of anti-VEGF agent.

## 2. Materials and Methods

A prospective interventional study carried on patients with exudative form of AMD who underwent multiple intravitreal injections of bevacizumab (IVB) at Ophthalmology Eye Hospital “Clinic Maja”, Nis, Serbia. IVB were given in standard dose (1.25 mg in 0.05 mL), according to the PRN protocol. Exudative AMD was defined as intra/subretinal or sub–RPE hemorrhages and/or fluid with or without fibrosis proved by OCT and FAG. OCT scans were obtained using Cirrus™ (Carl Zeiss Meditec Inc, Dublin, OH, USA) spectral domain optical coherence tomography (SD-OCT). Thickness of peripapillary RNFL (average and fields: superior, nasal, inferior, and temporal) and macular GC + IPL (the average, minimum and in six sectors: superior- temporal, superior, superior-nasal, inferior-nasal, inferior, inferior-temporal). GCL + IPL thicknesses were measured from the elliptical annulus centered on the fovea. Minimum GCL + IPL thickness parameter given as a value on the printout of OCT represents the minimum thickness acquired within the area of measurement. Demarcation of superior sectors was performed from nasal to temporal, thus the superior-nasal sector of GCL + IPL was marked as GC1, superior as GC2, superior-temporal as GC3, while demarcation of inferior sectors was performed from temporal to nasal, marking inferior-temporal as GC4, inferior GC5, and inferior-nasal as GC6.

All patients underwent complete ophthalmological examination prior the first anti-VEGF injection and followed up for 24 months in regular post-treatment intervals (after 6 months, 1 year, and 2 years). Exclusion criteria were: presence of any ocular disease that could affect RNFL and GCL + IPL thickness, particularly glaucoma and neurologic diseases; previous ocular surgeries; and laser interventions. Furthermore, in order to prevent algorithm segmentation failure in measuring RNFL thickness, OCT images with missing parts, misplacement of boundaries between retinal layers, or seemingly distorted images that resulted in readings of zero or otherwise abnormally low values were excluded. Patients who experienced any cerebro-vascular accident or myocardial infarction were excluded as well.

The study has followed the tenets of the Declaration of Helsinki. All patients signed informed consent prior to each intravitreal injection.

Data were analyzed using IBM SPSS Statistics for Windows, Version 22.0 (IBM Corp, Armonk, NY, USA). Primary obtained data were analyzed by descriptive statistical methods and methods for hypotheses testing. Descriptive statistical methods used were measures of central tendency (arithmetic average, median) and rate variability (standard deviation and variation interval). To determine the normality of distribution the coefficient of variation (CV) is used, Kolmogorov–Smirnov and Shapiro–Wilk testing as well. To test the arithmetic average differences between groups independent samples *t*-test was used. The variation of numerical data (GC1, GC2, GC3, GC4, GC5, GC6, average and minimum GCL + IPL thickness, superior, nasal, inferior, temporal, and average RNFL thickness) was assessed using general linear models with repeated-measures ANOVA. Pairwise comparisons were Bonferroni adjusted. The level of statistical significance was 0.05.

## 3. Results

In total, 135 consecutive patients with exudative AMD (70 women and 65 men) were included in this study. The mean age of patients was 65 ± 15 years. The average number of injections per patient was 12.43 ± 2.4. The average RNFL thickness prior the first injection was 87.64 ± 12.23 and after two years 86.23 ± 12.55 (*p* > 0.05) (Table 1).

Average GC + IPL thickness prior to the first IVB was 47.23 ± 15.45, and after two years 46.67 ± 11.87 (*p* > 0.05) (Table 2). Average RNFL thickness, average GC + IPL thickness, and all sectorial RNFL thickness values before anti-VEGF administration and after two-year follow-up did not differ significantly (*p* > 0.05). However, during the study period, significant decreases were observed in thickness values in GC2, GC5 segments, and minimum GCL + IPL (Table 2).

## 4. Discussion

The effect of multiple intravitreal anti-VEGF injections on peripapillary RNFL and macular GC + IPL thickness was investigated in 135 patients with exudative form of AMD. This group of patients is reasonably large when compared to other studies [13]. The average number of injections per patient was fairly high 12.4 ± 2.4 and comparable to other studies [14,15,16,17]. Recorded values of average RNFL thickness and segmental RNFL thickness prior to the first injection and after two-years of follow-up did not differ significantly (*p* > 0.05). Similar results were found in eight studies which analyzed long-term changes in RNFL thickness after multiple anti-VEGF treatment: either in the same eye during a period of one year [3] or longer [16], or compared treated eye with the fellow eye with dry AMD [14,18,19,20,21], or with patients with untreated AMD [22], or with healthy subjects [14]. Furthermore, Demirel et al. [14] and Shin et al. [22] in a long-term follow-up found no correlation between total number of anti-VEGF injections received and RNFL thickness [14]. Shin et al., compared effects of anti-VEGF treatment in different ocular conditions (including AMD) and concluded that loss of RNFL thickness was observed only in diabetic retinopathy and retinal vein occlusion concluding that retinal ischemia could be the main cause of RNFL loss rather than anti-VEGF treatment [22].

Luke et al., analyzed OCT parameters (such as total macular volume, RNFL, and central retinal thickness) in 120 eyes treated with multiple intravitreal injections of ranibizumab, bevacizumab, and aflibercept due to neovascular AMD in a clinical real world setting study. They found out that RNFL remained constant over the course of 8 years [23]. Finally, meta-analysis conducted by Shin et al., found no association between anti-VEGF treatment and RNFL thickness changes in AMD patients [24]. Wang et al., analyzed RNFL thickness in 54 eyes treated by multiple anti-VEGF injections and found that the relationship between the number of injections and RNFL thickness became apparent after approximately 30 injections and 50 months of the treatment course [25].

However, in studies of both Martinez-de-la-Casa et al. [13] and Beck et al. [26], RNFL thickness significantly decreased in the treated eye while no changes were observed in the control (untreated) eye without exudative AMD during the long-term follow-up. Furthermore, Beck et al., found that the presence of RPE atrophy negatively correlated with RNFL thickness [26]. Enders et al. [15] hypothesized that RNFL loss is caused by the IOP elevation after intravitreal injection and compared patients which along with anti-VEGF application had anterior chamber (AC) paracentesis compared to those without this additional intervention and found that patients without paracentesis had significantly higher incidence of RNFL loss, while AC paranetesis prevented the RNFL loss. Gomez-Mariscal et al. [27] evaluated acute and chronic changes in optic nerve head (ONH) in 29 nAMD eyes receiving multiple anti-VEGF injections. They observed a significant Bruch’s membrane opening (BMO), enlargement, cup widening and deepening, transient increase in intraocular pressure and prelaminar tissue thinning in the first 5 min after each injection. After one year, in 13 eyes receiving more than six anti-VEGF injections, significant and irreversible changes occurred—such as BMO expansion, prelaminar tissue thinning, and cup deepening in inferior region of the ONH.

Parlak et al., found that RNFL thickness decreased both in the treated and untreated eye during the study period and proposed that AMD alone could be associated with progressive RNFL loss [20]. Zucchiatti et al., investigated OCT findings in different stages of AMD and documented significantly reduced RNFL (as well as macular GC + IPL) thickness in untreated neovascular AMD when compared to healthy control subjects and concluded that reduction in RNFL and GC + IPL thickness could be part of clinical presentation of the disease [28]. Jo et al. [18] compared different segments of RNFL corresponding to the healthy unaffected retina and to the affected macular regions and concluded that RNFL thickness significantly decreased in pathologic areas and that progression of macular disease, rather than anti-VEGF treatment, affected segmental RNFL thickness. However, Rimayanti et al., found no difference in RNFL thickness between patients with untreated neovascular AMD and healthy controls (except of patients with AMD and glaucoma where RNFL thickness was reduced) [29]. Similarly, Valverde-Megias et al. [30] found the decrease in RNFL equally in the injected (with nAMD) and control (dry AMD) group and concluded that ranibizumab had no long-term effect on RNFL thickness. They observed that RNFL thickness in superior temporal sector demonstrated the greatest loss in both groups, followed by inferior and nasal sectors. Further studies are needed to determine the clinical course of neovascular AMD and long-term consequences of anti-VEGF treatment on RNFL thickness in AMD patients.

In our study, although values of average macular GC + IPL thickness did not change during follow-up, a significant decrease was observed in GC2, GC5 segments, and in minimum GC + IPL thickness. Only a few studies have analyzed ganglion cell and inner plexiform layer thickness after repeated treatment with anti-VEGF. Zucchiatti et al., found no difference in ganglion cell complex (GCC) thickness in patients with exudative AMD during one-year follow-up. However, they detected reduced choroidal thickness in papillomacular area and reduced central macular thickness [28]. On the other hand, Saleh et al. [17] had average follow-up of 6 years and found significantly reduced average macular retinal ganglion cell layer (GCL) thickness and this reduction correlated with number of intravitreal injections. However, both studies involved rather small number of patients (24 and 16, respectively), so additional larger studies are needed in particularly due to the fact that loss of GCC could be part of the natural course of the exudative AMD, as mentioned earlier.

Demir et al. [31] compared RNFL, GCL, IPL, inner nuclear leyer thickness in 25 eyes with nAMD with 25 dry AMD eyes, as well as thickness of lesion-free retina in the outer nasal subfield of ETDRS grid during the one year of follow-up. They concluded that the retinal thickness decreased during the first three injections in nAMD group. Repeated anti-VEGF injections had no influence on RNFL, GCL, IPL, and INL thickness of the lesion-free retina. Even the GCL became thinner in the study group, it was not significantly up to as of the end of one year period. That is similar result to ours even though it is not fully comparable in parameters, since we analyzed GCL + IPL thickness. Similar to our study, Lee et al. [32] analyzed GC + IPL thickness but they compared groups with nAMD eyes which received ranibizumab, aflibercept, or combinations of both anti-VEGF and found out that GC + IPL thickness decreased in aflibercept and combined group, while RNFL decreased in ranibizumab and combined group 23.

In the future, a larger multi-centric study with a longer follow-up period (at least five years) are needed to further analyze the long term effects of different anti-VEGF agents of retinal tissue in patients with exudative AMD.

## 5. Conclusions

Repeated anti-VEGF intravitreal application did not cause significant thinning of average RNFL and GC + IPL thickness over a period of 24 months following the initial dose, but there was a trend towards thinning of GC + IPL thickness in some sectors after two years. These findings could be a sign of ganglion cell damage induced by anti-VEGF treatment or occurring during the natural course of the disease. Patients suffering from glaucoma were excluded from the study, so results cannot be applied to patients with glaucoma receiving intravitreal anti VEGF injections.

## Figures and Tables

**Table 1 medicina-59-00138-t001:** Mean values of RNFL thickness in group of patients that received multiple anti-VEGF intravitreal injections during a two-year follow-up.

RNFL Thickness (µm)	Initial	After 6 Months	After 1 Year	After 2 Years	*p*-Value
Superior RNFL *	99.6 ± 17.8	99.8 ± 19.0	101.0 ± 18.7	101.6 ± 17.1	0.234
Nasal RNFL *	70.9 ± 10.5	70.8 ± 11.3	71.5 ± 10.7	69.0 ± 13.0	0.402
Inferior RNFL *	112.3 ± 20.5	113.5 ± 21.7	112.9 ± 20.7	113.6 ± 20.8	0.063
Temporal RNFL *	64.1 ± 11.6	64.0 ± 13.4	65.0 ± 13.4	64.3 ± 10.4	0.092
Average RNFL *	87.64 ± 12.23	87.6 ± 10.7	87.0 ± 11.0	86.23 ± 12.55	0.126

RNFL: retinal nerve fiber layer; *: Mean value ± standard deviation.

**Table 2 medicina-59-00138-t002:** Mean values of ganglion cell values in group of patients that received multiple anti-VEGF intravitreal injections during a two year follow up.

GC Thickness (µm)	Initial	After 6 Months	After 1 Year	After2 Years	*p*-Value
GC1 *	47.5 ± 12.8	47.2 ± 13.5	47.1 ± 13.2	47.0 ± 12.8	0.219
GC2 *	46.8 ± 11.8	45.7 ± 12.0	45.6 ± 12.8	43.9 ± 12.4	**0.037**
GC3 *	47.5 ± 13.2	47.5 ± 13.8	47.8 ± 13.6	47.8 ± 14.2	0.062
GC4 *	48.3 ± 12.7	47.5 ± 13.1	48.0 ± 12.4	47.7 ± 13.0	0.119
GC5 *	44.4 ± 12.6	43.3 ± 13.2	42.5 ± 13.2	41.9 ± 13.4	**0.044**
GC6 *	44.0 ± 12.3	43.7 ± 13.1	44.4 ± 13.2	41.4 ± 12.2	0.073
GCL + IPL Average *	47.23 ± 15.45	47.4 ± 7.2	47.67 ± 7.4	46.67 ± 11.87	**0.077**
GCL + IPL Minimum *	41.5 ± 9.1	40.3 ± 9.7	41.5 ± 10.0	39.0 ± 10.0	0.017

GC1—supero-nasal sector of ganglion cell-inner plexiform layer; GC2—superior sector of ganglion cell-inner plexiform layer; GC3—supero-temporal sector of ganglion cell-inner plexiform layer; GC4—infero-temporal sector of ganglion cell-inner plexiform layer; GC5—inferior sector of ganglion cell-inner plexiform layer; GC6—infero-nasal sector of ganglion cell-inner plexiform layer; GCL + IPL Average—ganglion cell-inner plexiform layer average thickness; GCL + IPL Minimum—ganglion cell-inner plexiform layer minimum thickness; *: mean value ± standard deviation. Note: bold *p* values are those which are statistically significant.

## Data Availability

Data supporting reported results can be sent if needed or requested by the reviewer.

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
