# Peer review of "Influence of Multiple Anti-VEGF Injections on Retinal Nerve Fiber Layer and Ganglion Cell-Inner Plexiform Layer Thickness in Patients with Exudative Age-Related Macular Degeneration"

_medicina, 2023, doi:10.3390/medicina59010138_

Round 1

Reviewer 1 Report

December 4th, 2022

Influence of multiple anti-VEGF injections on retinal nerve fiber layer and ganglion cell-inner plexiform layer thickness in 3 patients with exudative age-related macular degeneration.

Anti-VEGF therapy has improved functional outcome for many patients with neovascular AMD. A particular challenge in routine clinical application is to find the best treatment regimen as a high degree of interindividual variability of disease activity. Objective of the article: To analyze the influence of multiple anti-VEGF intravitreal injections for exudative age-related macular degeneration on the thickness of peripapillary retinal nerve fiber layer (RNFL) and macular ganglion cell-inner plexiform layer (GC+IPL) using spectral domain optical coherence tomography (SD-OCT).

The study provides valuable clinical information regarding disease progression over 2 years in 135 patients undergoing multiple intravitreal injections using bevacizumab. For this, different areas of the retina in a patient with exudative AMD were evaluated, including: thickness of peripapillary RNFL (average and fields: superior, nasal, inferior and temporal) and macular GC+IPL (the average, minimum and in six sectors: superior - temporal, superior, superior-nasal, inferior-nasal, inferior, inferior-temporal). The prospective intervention concludes that repeated doses of anti-VEGF intravitreal application do not cause a significant decrease in the average thickness of RNFL and GC+IPL. These results are very well discussed with papers that have reported similar behavior and other studies that have shown beneficial effects.

However, the work has some limitations such as not presenting imaging studies based on fundus photography and fluorescein angiography however, these methods only detect structural sequelae, and the sensitivity is lower than that of molecular imaging technology. The confocal scanning laser ophthalmoscope (cSLO) is a common instrument used for fluorescein angiography and fundus autofluorescence and this instrument can be used for in vivo molecular imaging of VEGF expression.

In relation to these observations, I have some questions:

1) What differences and contributions to disease progression can thickness of peripapillary RNFL and macular GC+IPL have?

2) Can you explain the relationship between in vivo imaging (fundus photography, fluorescein angiography and optical coherence tomography) and the pharmacokinetics of VEGF?

3) Some clinical markers are rather nonspecific and primarily address anatomic/morphologic features, while they are not able to detect preceding alterations on a molecular level. In this aspect, what do you suggest?

4) Why didn't they evaluate other doses of anti-VEGF to compare against the therapeutic dose of 2.5mg

Author Response

Concerning to:

Manuscript ID: medicina-2062926

Title

Influence of multiple anti-VEGF injections on retinal nerve fiber layer and ganglion cell- inner plexiform layer thickness in patients with exudative age-related macular degenerationReviewer 1:

Dear Sir/Madam,

thank you for the useful comments and very interesting questions that made us think about new ideas for future projects

Authors (22/12/2022)

Response

Reviewer 1: Influence of multiple anti-VEGF injections on retinal nerve fiber layer and ganglion cell-inner plexiform layer thickness in 3 patients with exudative age-related macular degeneration.

Comment by authors: Influence of multiple anti-VEGF injections on retinal nerve fiber layer and ganglion cell-inner plexiform layer thickness in 135 patients with exudative age-related macular degeneration

Reviewer 1: The study provides valuable clinical information regarding disease progression over 2 years in 135 patients undergoing multiple intravitreal injections using bevacizumab. For this, different areas of the retina in a patient with exudative AMD were evaluated, including: thickness of peripapillary RNFL (average and fields: superior, nasal, inferior and temporal) and macular GC+IPL (the average, minimum and in six sectors: superior - temporal, superior, superior-nasal, inferior-nasal, inferior, inferior-temporal). The prospective intervention concludes that repeated doses of anti-VEGF intravitreal application do not cause a significant decrease in the average thickness of RNFL and GC+IPL. These results are very well discussed with papers that have reported similar behavior and other studies that have shown beneficial effects.

Comment by authors: thank you for this comment. There are few studies related to this topic, so we did a very detailed literature research

Reviewer 1: However, the work has some limitations such as not presenting imaging studies based on fundus photography and fluorescein angiography however, these methods only detect structural sequelae, and the sensitivity is lower than that of molecular imaging technology. The confocal scanning laser ophthalmoscope (cSLO) is a common instrument used for fluorescein angiography and fundus autofluorescence and this instrument can be used for in vivo molecular imaging of VEGF expression.

Answer: definitely, cSLO can be utilized for in vivo molecular imaging of VEGF expression though mostly it is available in experimental models.  As Zhang L. et al. (Biomed Opt Express. 2021 Oct 29;12(11):7185-7198) stated, the structure of the eye provides a unique platform for light-based molecular imaging. Also, high sensitivity and specificity of the molecular imaging probes make them highly suitable for the detection of VEGF expression.  Several applications of in vivo molecular imaging of retinal diseases have been reported. For example, Cordeiro et al. studied retinal nerve cell apoptosis in vivo with fluorescent-labeled annexing 5 probes. Furthermore, Sun et al. revealed increased expression of vascular endothelial growth factor receptor 2 in the retinal capillary endothelia of diabetic rats compared with controls in vivo

In relation to these observations, I have some questions:

  • What differences and contributions to disease progression can thickness of peripapillary RNFL and macular GC+IPL have?

Answer: In our opinion mentioned layers are not directly involved in the progression of AMD nor do they affect the final outcome of the underlying disease. These layers "suffer" as a result of neovascular membrane activity and exudation and, in later stages of the disease, due to fibrosis. So, we do not think that they affect AMD progression or cause the outcome of AMD to differ depending on which layer is affected. However, hypothetically, these layers can be changed under the influence of antiVEGF injections, which was the aim of our study

  • Can you explain the relationship between in vivo imaging (fundus photography, fluorescein angiography and optical coherence tomography) and the pharmacokinetics of VEGF?

Answer: relation between in vivo imaging (fundus photography, fluorescein angiography and optical coherence tomography) and the pharmacokinetics of VEGF have not been examined in AMD, according to our research and up-to-date data. However, some results have been published regarding diabetic retinopathy in experimental model (Zhang L. et al. In vivo fluorescence molecular imaging of the vascular endothelial growth factor in rats with early diabetic retinopathy. Biomed Opt Express. 2021 Oct 29;12(11):7185-7198)

  • Some clinical markers are rather nonspecific and primarily address anatomic/morphologic features, while they are not able to detect preceding alterations on a molecular level. In this aspect, what do you suggest?

Answer: thank you for this comment and question. Regarding to clinical markers, whatever we observe, changes due to AMD or influence of antiVEGF injections, we are almost always focused on morphologic features. On molecular level, in this moment, it is impossible to detect any alteration, except in experimental models. But we hope it will become possible in near future.

  • Why didn't they evaluate other doses of anti-VEGF to compare against the therapeutic dose of 2.5mg

Answer: we evaluate regular therapeutic dose of 2.5 mg since it is everyday clinical practice.  But, it is a good idea for some further investigations.

Reviewer 2 Report

Maja et al. studied the effects of multiple anti-VEGF intravitreal injections on retinal nerve fiber layer thickness and ganglion cell-inner plexiform layer thickness. The authors have found that GC+IPL thickness was decreased by the treatment.

There are a few major concerns:

1.       The references are not in order. Kindly check.

2.       Were these patients’ diabetic history considered? Kindly provide the details.

3.       If there are already many studies that evaluated the effects of multiple anti-VEGF injections on RNFL and GC+IPL thickness, what is the importance of the present study? Kindly highlight the novelty of the study.

Author Response

To the Reviewer 2:

thank you very much for the useful comments and very interesting questions. Authors

Response to the reviewer 2  (22/12/2022)

Concerning to: Manuscript ID: medicina-2062926

Title: Influence of multiple anti-VEGF injections on retinal nerve fiber layer and ganglion cell- inner plexiform layer thickness in patients with exudative age-related macular degeneration

Dear Sir/Madam,

thank you for the useful comments and very interesting questions that made us be better

Authors (22/12/2022)

There are a few major concerns:

  1. The references are not in order. Kindly check.

Comment: when we received the preprint of our paper, we were surprised that the references were "out of order" which was not the case when we submitted the paper to the journal. We wrote an email and received a reply that the current technical error is being worked on and that the editors will fix it. We will certainly insist again to do it. Thank you for this comment.

  1. Were these patients’ diabetic history considered? Kindly provide the details.

Answer: yes, diabetic history was considered but it was not exclusion criteria except in the cases of proved diabetic retinopathy in any stage (pg2, lines 89-92). At the end, in study group, there were not diabetic patients

  1. If there are already many studies that evaluated the effects of multiple anti-VEGF injections on RNFL and GC+IPL thickness, what is the importance of the present study? Kindly highlight the novelty of the study.

Answer: we did not find many studies focused on GCIPL measurement in the cases of multiple intravitreal injections due to AMD. The reason could be in the fact that only a few OCT machine could provide it. Having on mind that ganglion-cell-inner-plexiform layer (GCIPL) and RNFL constitute 30–35% of the total macular thickness and the loosing of RGC may be detectable a long before obvious functional deficit.  We hypothesized that there are some changes in GCIPL much before we “see” it clinically.

Novelty of this study were the following:

  • we found decrease in specific GCIPL sectors
  • however, studies which we found, involved significantly smaller number of patients (up to 24) compare to our study (135)
  • follow up period was two years comparing to the other studies, it is a much longer

Round 2

Reviewer 2 Report

This paper can be accepted in the present form.